# Lipoarabinomannan from *Mycobacterium indicus pranii* shows immunostimulatory activity and induces autophagy in macrophages

Bindu Singh[¤a☯], Mohd Saqib[¤b☯], Anush Chakraborty, Sangeeta Bhaskar[iD]*

Product Development Cell-1, National Institute of Immunology, New Delhi, India

☯ These authors contributed equally to this work.
¤a Current address: Texas Biomedical Research Institute, San Antonio, Texas, United States of America
¤b Current address: Albany Medical Center, Albany, New York, United States of America
* sangeeta@nii.ac.in

**Data Availability Statement:** All relevant data are within the manuscript and its Supporting Information files.

## Abstract

*Mycobacterium indicus pranii* (MIP) known for its immunotherapeutic potential against leprosy and tuberculosis is undergoing various clinical trials and also simultaneously being studied in animal models to get insight into the mechanistic details contributing to its protective efficacy as a vaccine candidate. Studies have shown potential immunomodulatory properties of MIP, the most significant being the ability to induce strong Th1 type of response, enhanced expression of pro-inflammatory cytokines, activation of APCs and lymphocytes, elicitation of *M.tb* specific poly-functional T cells. All of these form crucial components of host-immune response during *M.tb* infection. Also, MIP was found to be potent inducer of autophagy in macrophages which resulted in enhanced clearance of *M.tb* from MIP and *M.tb* co-infected cells. Hence, we further examined the component/s of MIP responsible for autophagy induction. Interestingly, we found that MIP lipids and DNA were able to induce autophagy but not the protein fraction. LAM being one of the crucial components of mycobacterial cell-wall lipids and possessing the ability of immunomodulation; we isolated LAM from MIP and did a comparative study with *M.tb*-LAM. Stimulation with MIP-LAM resulted in significantly high secretion of pro-inflammatory cytokines and displayed high autophagy inducing potential in macrophages as compared to *M.tb*-LAM. Treatment with MIP-LAM enhanced the co-localization of *M.tb* within the phago-lysosomes and increased the clearance of *M.tb* from the infected macrophages. This study describes LAM to be a crucial component of MIP which has significant contribution to its immunotherapeutic efficacy against TB.

## Introduction

*Mycobacterium tuberculosis (M.tb)*, the causative agent of tuberculosis (TB), poses a serious health threat to the public worldwide. *M.tb* is an intracellular pathogen residing within the

**Funding:** This work was supported by the Core Research Grant of the National Institute of Immunology.

**Competing interests:** The authors have declared that no competing interests exist.

**Abbreviations:** BSA, Bovine serum albumin; CFU, Colony forming units; DAPI, 4′,6-Diamidine-2′-phenylindole dihydrochloride; H$_2$DCFDA, 2′,7′-dichlorodihydrofluorescein diacetate; DMEM, Dulbecco's Modified Eagle Media; LAM, Lipoarabinomannan; MIP, *Mycobacterium indicus pranii*; *M.tb, Mycobacterium tuberculosis*; PBS, Phosphate buffered saline; LC3, Microtubule-associated proteins 1A/1B light chain 3B; PFA, Paraformaldehyde; PVDF, Polyvinylidenedifluoride; RPMI, Roswell Park Memorial Institute medium; TB, Tuberculosis.

mononuclear phagocytes, has developed specific mechanisms to evade the host innate immune response which facilitate its long-term survival [1,2]. Phagosome maturation and phago-lysosome fusion block, interference with antigen presentation, resistance to reactive oxygen and nitrogen intermediates [3,4], alteration of host cell apoptotic pathways [5] and inhibition of autophagy in host cells [6] count among the strategies which enhance *M.tb* survival inside the macrophages.

A multitude of innate immune signaling pathways are involved in host defense against pathogens. During infection, the pathogen-associated molecular patterns of the microbes are recognized by host cells through various pattern recognition receptors. This recognition provokes an intracellular signaling cascade, resulting in activation of antimicrobial effector mechanisms to stimulate the clearance of the pathogens [7,8]. Autophagy works as one of the effector mechanism downstream to these receptors and owing to this reason it forms an integral part of innate and adaptive immunity to various pathogens [9]. Recent reports have demonstrated that autophagy induction in macrophages plays a crucial role in the innate immune response to *M.tb* [10].

Previously, we have reported that MIP is a potent inducer of autophagy in macrophages which resulted in enhanced co-localization of *M.tb* as well as its clearance from the infected macrophages [11]. The next question asked was whether MIP induced autophagy was exclusively by active mechanisms i.e. presence of whole MIP is required or some of its component/s have the ability to induce autophagy. Reports suggest that mycobacterial gene products / individual fractions can affect or limit host autophagy responses to the pathogens [12–15]. The protein/lipid/DNA fractions of MIP were isolated and tested for their ability to modulate autophagy in RAW 264.7 macrophages. MIP lipid as well as DNA fraction was able to induce significant autophagic response in macrophages. LAM is considered to be one of the prominent components of the lipid fraction with an established immunomodulatory potential. LAM from pathogenic mycobacteria has been reported to impose block in the autophagic pathway and also limit the fusion of phagosomes with the lysosomes [16–18]; thus help the bacilli to escape host immune mechanisms and enhance their survival inside the macrophages. Furthermore, MIP is known to be non-pathogenic mycobacteria and its autophagy inducing potential is known, we speculated the possibility that LAM might play crucial role in inducing autophagy in macrophages.

To test this hypothesis, MIP-LAM was isolated and purified and analyzed for its immunostimulatory as well as autophagy modulating properties. MIP-LAM led to significant production of pro-inflammatory cytokines including TNF-α, IL-6 and IL-12. Also, MIP-LAM was able to induce autophagy in macrophages. Enhanced co-localization of *M.tb* within the phago-lysosomes was observed in MIP-LAM stimulated macrophages which resulted in the increase in *M.tb* killing.

## Materials and methods

### Reagents and antibodies

RAW 264.7 (TIB-71$^{TM}$) and J774.1macrophages (TIB-67$^{TM}$) were obtained from ATCC. Roswell Park Memorial Institute medium-1640 (RPMI-1640) and Dulbecco's Modified Eagle Medium (DMEM) and Penicillin-Streptomycin antibiotic mixture were purchased from HiMedia Laboratories (Mumbai, India). Heat inactivated fetal bovine serum (FBS) was procured from Biological Industries (Israel). C57BL/6 mice were provided by small animal facility of National Institute of Immunology. Middlebrook 7H9 and Middlebrook 7H11-agar were obtained from Difco Laboratories, USA. MIP-LAM was prepared by us whereas *M.tb*-LAM (Cat. No. NR-14848), and monoclonal anti-LAM antibodies: CS-35 (Cat. No. NR-13811), and

CS-40 (Cat. No. NR-13812) were procured from BEI resources (Virginia, USA). ELISA kits for TNF-α (Cat. No. 555268), IL-6 (Cat. No. 555240) and IL-12 (Cat. No. 555165) were procured from BD Biosciences (California, USA). Recombinant murine Granulocyte-Macrophage Colony-Stimulating Factor (GM-CSF; Cat. No. 315–03) was purchased from Peprotech Asia (Rehovot, Israel). Pierce BCA protein assay kit (Cat. No. 23252), DQ-BSA (Cat. No. D12051), Lysotracker-Red (Cat. No. L7528), 4, 6-Diamidino-2-phenylindole (DAPI; Cat. No. D1306), ProLong Gold antifade reagent (Cat. No. P36934) were obtained from Invitrogen, Thermo Fisher Scientific (Waltham, NY, USA). Hygromycin-B (Cat. No. H274), Clarity Western ECL substrate (Cat. No. 170–5061), BLUelf Prestained Protein Ladder (Cat. No. PM008), were respectively obtained from Sigma Aldrich, Merck (Darmstadt, Germany), BioRad Laboratories (Hercules, California, USA); Bio-Helix Co. Ltd. (Keelung, Taiwan). LC3-II/I antibody (Cat. No. 12741), β-actin (Cat.No. 5125) and horseradish peroxidase (HRP)-conjugated goat anti-rabbit IgG antibody (Cat. No. 7074) and Alexa fluor 488 mAb were obtained from Cell Signaling Technology (Danvers, MA).

## Mycobacterial culture

MIP and *M.tb* (H37Rv strain) were cultivated in Middlebrook 7H9 broth supplemented with 10% ADC at 100 rpm in 37°C. Plating was done on Middlebrook 7H11-OADC plates containing Hygromycin B (conc. 2 μg/ ml). Oleic acid, Catalase, Lowenstein-Jensen (LJ) Media were obtained from HiMedia Laboratories, India. GFP expressing *M.tb* was prepared by the method described in [11].

## Cell culture

RAW 264.7 and J774.1 macrophages were cultured in RPMI-1640 and DMEM respectively, supplemented with 10% FBS and 1 X antibiotic cocktail. For experiments, macrophages were harvested by centrifugation at 300 g for 10 min and counted using hemocytometer. Suitable dilution was prepared and cells were seeded and kept overnight at 37°C in a $CO_2$ incubator for adherence.

## Extraction of whole lipid fraction of MIP

MIP culture ($OD_{600}$ = 0.8–0.9) was harvested by centrifugation at 1000 g for 10 min. The cell pellet was resuspended in chloroform-methanol-PBS solution in the ratio of 8:4:3 and allowed to mix by stirring for 4–5 h. The contents were then left stagnant for 4–5 h at RT to allow the separation of chloroform-soluble fraction from the methanol-soluble fraction. The chloroform layer containing lipids was then separated and the solvent was evaporated using BUCHI Rotavapor R-200. The dried lipids were then collected, weighed and were stored at -20°C till further use.

## Isolation of MIP genomic DNA

MIP genomic DNA isolation was done as per the protocol described in [19]. Briefly, 20–40 mL of MIP culture was harvested and the pellet was resuspended in 10 mL of Tris-EDTA buffer (TE buffer) and mixed gently. The mycobacterial suspension was heated at 80°C for 1 h in a water bath. The temperature of the suspension was lowered to 30°C and Lysozyme (final concentration 2 mg/ml) was added. The suspension was then incubated overnight at 37°C and thereafter, 1 ml of 10% SDS solution and 20 μl of 10 mg/ml Proteinase K solution (Final concentration- 20 μg/ml) was added. Equal volume of PCI mixture (Phenol:Chloroform:Isoamylalcohol in ratio of 25:24:1 v/v) was added and mixed properly. The resulting suspension was

then centrifuged at 12,000 g for 10 min at 4˚C. The aqueous phase was collected and 100 μg of 3 M sodium acetate solution per ml of aqueous phase was added to it and mixed well. Equal volume of Isopropanol was added and kept at stagnant position for 5–10 min. The suspension containing precipitated DNA was again centrifuged at 12,000 g for 10 min at 4˚C. Subsequently, the pellet was washed with 75% ethanol and air dried. Finally, the pellet was dissolved in molecular grade water and was analyzed by agarose gel electrophoresis using Bio Rad Gel Doc XR$^+$ System.

## Extraction of total MIP proteins

MIP culture was harvested and washed with PBS followed by lysis for 30 min on ice using lysis buffer. The lysed mycobacterial cells were sonicated 3 times for 20 sec at 50% power using a QSonica Ultrasonic Digital Sonicator to release protein followed by centrifugation at 11,000 rpm for 20 min at 4˚C. The supernatant was collected and concentrated using an Amicon Ultra Centrifugal Filter with molecular weight cut-off of 10,000 kDa until a concentration of 1–2 mg/ml was achieved. The protein sample was confirmed using SDS PAGE and subsequently protein concentrations were determined using BCA assay.

## LC3 Western blotting

The cellular lysates were prepared using M-2 lysis buffer (1 M Tris pH 7.4; 5 M NaCl; Glycerol; 10% Triton-X-100; 0.5 M EDTA; 0.5 M EGTA) containing 1 X PIC (Protease Inhibitor Cocktail). The protein content of cell lysates was determined using BCA method of protein estimation and expression of LC3 was evaluated using Western blotting. Briefly, 20 μg of each protein sample was loaded on 15% SDS gel and electrophoresis was done. The proteins on the gel were then transferred onto Polyvinylidenedifluoride (PVDF) membrane using wet transfer at 60 V for 2.5 h. Blocking was done at RT for 1 h using 5% w/v skimmed milk prepared in 1 X TBST (Tris-buffered saline with 0.1% Tween-20). Subsequently membrane was incubated with LC3 II/I primary antibody (1:1000 dilution in 5% w/v Bovine Serum Albumin in TBST) for overnight at 4˚C. This was followed by incubating the membrane for 1 h with HRP tagged secondary antibody (1:5000 dilution in TBST) at RT. Thereafter, the proteins were detected by chemiluminescence and images were analyzed using ImageJ software.

## Purification of MIP lipoarabinomannan

Cell wall was prepared by method followed by our group [20]. Briefly, MIP culture in mid log growth phase was harvested by centrifugation at 3000 g for 10 min. The pellet was washed with PBS, resuspended in cold PBS and passed through French press twice at 40,000 kPa. The resulting suspension was centrifuged at 10,000 g for 15 min and the pellet was discarded. Supernatant was again centrifuged at 27,000 g for 30 min. The pellet consisting of cell wall was lyophilized. The dry cell wall was re-hydrated in PBS (10 g in 100 ml PBS) and the resulting suspension was sonicated at 40% efficiency for 15 min. The suspension was then centrifuged at 18,000 g for 30 min and supernatant was collected which was subsequently concentrated by vacuum evaporation. The concentrated supernatant was extracted with phenol (40% final concentration) for 1 h at 70˚C. Aqueous phase was separated from phenol layer by low speed centrifugation (4000 g for 20 min) and phenol phase was extracted once more with water. The aqueous extracts from both extractions were combined and concentrated by vacuum evaporation to 50 ml final volume. This concentrated aqueous extract was extracted with four volumes of chloroform: methanol (2:1) for the removal of residual phenol. The aqueous phase having crude LAM was collected and evaporated to dryness. Crude LAM was re-suspended in PBS (10 mg in 1 ml PBS) and passed through sephacryl S-100 column equilibrated with PBS at flow

rate of 0.5 ml/min and 3 ml fractions were collected. Each fraction was examined for carbohydrate and protein content by phenol-sulphuric acid method and BCA respectively.

Presence of LAM was detected by ELISA using reference mAbs CS-35 and CS-40 (1:1 ratio at dilution of 1:500). Fractions found positive for presence of LAM were pooled and concentrated by using an Omega cell (10 kDa molecular cut-off membrane). Finally, LAM was precipitated by ethanol at final concentration of 80%.

## Carbohydrate content determination

Carbohydrate content in each fraction obtained from sephacryl-S100 column was determined by phenol-sulphuric acid method. 50μl of each fraction was taken in 1.5 ml tube and 167 μl phenol (4%) and 834 μl $H_2SO_4$ (96%) were added to it. Contents were shaken and incubated for 10–20 min at RT. Subsequently, O.D. was measured by ELISA reader at 490 nm. Glucose was used as standard control.

## ELISA for identification of LAM containing fractions

96-well ELISA plate was coated with 100 μl of each fraction overnight at RT. *M.tb*-LAM was used as standard. Plate was washed 4 times with PBST (Phosphate buffer saline + 0.05% Tween 20). Blocking was done using 1% BSA for 1 h at 37˚C. Plate was washed thrice and 100 μl of mAbs CS-35 and CS-40 (dilution 1:1000 in PBS) were added, incubated for 1 h at RT. Plate was again washed and 100 μl of rabbit anti-mouse IgG-HRP conjugate (dilution 1:2000 in PBS) was added to the wells and incubated for 1 h at RT. Washing was done 5 times and subsequently 100 μl of TMB substrate was added to the wells. O.D. was measured at 450 nm by ELISA reader. Fractions found positive for LAM were pooled and concentrated by using an Omega cell (10 kDa molecular cut-off membrane). Finally, LAM was precipitated with ethanol (85% of total volume) and precipitate was collected, centrifuged and lyophilized.

## SDS-PAGE andsilver stainingof LAM

MIP-LAM was mixed with 1 X protein loading dye and heated at 95˚C for 5 min and desired quantity was loaded on to 10% SDS-PAGE and run at 30 mA for 90 min.

Silver staining of LAM requires modification of silver staining protocol used for visualization of proteins on gel. In this method, LAM resolved gel was treated with 0.2% periodic acid for 2 min at 40˚C before silver staining step. Gel was first placed in fixation solution overnight at 4˚C. Subsequently, it was dipped in oxidation solution for 5 min with gentle agitation followed by washing thrice with $ddH_2O$, each for 30 min. The gel was then transferred to a clean glass plate, staining solution was added to it and kept in shaking condition for 10 min. Gel was washed in $ddH_2O$ four times, each for 10 min with gentle agitation.

## Western blot of LAM

LAM sample resolved by SDS-PAGE was transferred on to polyvinylidenedifluoride membrane by diffusion method at 70˚C for 30 min in transfer buffer (39 mM glycine, 48 mM Tris, 0.037% SDS and 15% methanol). The membrane was blocked by 5% BSA in PBS for 1 h at 37˚C followed by three washes of 5 min each in PBST (PBS containing 0.05% tween-20). The membrane was then incubated with anti-LAM mAbs CS-35 and CS-40, diluted (1:1000) in PBST containing 5% BSA. The blot was again washed thrice with PBST and incubated for 1 h at RT with HRP conjugated anti-mouse secondary antibody (1:5000) prepared in 5% BSA-PBST. After three subsequent washes, the blot was developed using chemiluminescent substrate.

## Derivation of dendritic cells from bone-marrow

C57BL/6 mice (6–8 weeks old) were used for isolation of bone marrow dendritic cells (BMDCs). Femur and tibia from hind limb were removed aseptically and kept in complete RPMI medium after cleaning the muscular tissue. The bones were then washed for 2 min in 70% ethanol followed by two washes with RPMI medium. Bones were transferred to a sterile petri-dish and were cut open at both ends and subsequently flushed with 5 ml of complete RPMI medium using a syringe. Marrow plugs were passed from syringe once to break clumps and collected in a tube. Pellet of the marrow, obtained after centrifugation, was subjected to RBC lysis solution (3–4 ml Gey's solution for 3–5 min at RT) and washed twice at 300 g for 10 min. Cell density was adjusted to $1 \times 10^6$ cells/ml in complete RPMI, supplemented with 20 ng/ml (700–1000 U/ml) of murine recombinant Granulocyte-macrophage colony-stimulating factor (GM-CSF) and the suspension was plated (4 ml/well) in 6-well plate. Exhausted media was aspirated following gentle shaking of plate (to remove non-adherent granulocytes and lymphocytes) and replenished with fresh RPMI medium supplemented with mGM-CSF at day 3 and day 5. At day 7, cells were dislodged by gentle pipetting and pooled in a tube. The cells were washed twice to remove GM-CSF before setting up the assay.

## Analysis of immunomodulatory property of LAM

J774.1 macrophages or bone marrow derived dendritic cells were plated at respective densities of $0.5 \times 10^6$ or $1.0 \times 10^6$ per well in 24-well culture plates. Varying concentrations of MIP-LAM and *M.tb*-LAM were added to the wells and incubated for 24 h. Culture supernatant was collected and concentrations of cytokines (TNF-α, IL-12 and IL-6) were measured by ELISA using commercially available ELISA kits, according to the manufacturer instructions.

## NO and ROS estimation

RAW macrophages were seeded at a density of $1.0 \times 10^6$ per well in 12-well culture plates and stimulated with LPS (1 μg/mL) or MIP-LAM (5 μg/mL) for 24 h. Supernatant was collected and estimated for NO with Griess reagent.

For ROS estimation, the cell-permeant 2',7'-dichlorodihydrofluorescein diacetate (H$_2$DCFDA) was used. The non-fluorescent H$_2$DCFDA is converted to the highly fluorescent 2',7'-dichlorofluorescein (DCF) when the acetate groups are cleaved by intracellular esterases and oxidation. Briefly, macrophages were incubated with 20 μM H$_2$DCFDA dye for 45 min at 37°C under dark conditions. Cells were then washed followed by stimulation with LPS (1 μg/mL) / MIP-LAM (5 μg/mL). Fluorescence was determined using fluorimeter, excitation at 490 nm and emission at 524 nm. H$_2$O$_2$ (100 μM) was used as a positive control.

## Puncta formation assay

RAW 264.7 cells were seeded on sterile coverslips (18 x 18 cm$^2$) placed in 12-well plate and kept overnight for adherence. Next day, the macrophages were stimulated with varying concentrations of MIP-LAM or *M.tb*-LAM. After specified time-points, the cells were fixed using absolute methanol for 10 min. Permeabilization was done using 0.1% Triton-X-100 in PBS for 10 min at RT followed by blocking with 3% BSA (prepared in 0.1% Triton-X-100 in PBS) for 2 h at RT. Cells were then incubated overnight with anti-LC3 antibody (dilution-1:100) at 4°C and subsequent incubation with Alexa fluor 488 antibody (dilution-1:1000) for 45 to 60 min at RT. Slides were visualized using Zeiss LSM 510 Meta Confocal Microscope at 63 X magnification.

## Co-localization of *M.tb* with lysosomes in LAM stimulated macrophages

The protocol used in our previous publication was followed [11]. Briefly, macrophages were seeded on coverslips were infected with GFP expressing *M.tb* for 4 h with subsequent stimulation with MIP-LAM / *M.tb*-LAM for 4 h. This was followed by Lysotracker-Red staining. Fixation was done in 4% PFA for 10 min at RT with subsequent DAPI staining. Co-localization of *M.tb* expressing GFP with Lysotracker-red was examined by counting total number of green spots and yellow spots and % co-localization was calculated.

## Assessment of *M.tb* survival by CFU assay

1 x 10$^6$ macrophages were seeded per well in a 12-well plate and infected with *M.tb* for 4 h at a MOI of 1:10. After washing out the extracellular bacteria, cells were stimulated with MIP-LAM / *M.tb*-LAM for 4 h. Cells were then washed and lysed for 10 min using 0.1% Triton X-100 prepared in PBS. From each group, 100 μl of lysate was plated on 7H11-agar plates in triplicates. After 22–28 days, the colonies were counted and percentage survival of *M.tb* was quantified.

## Statistical analysis

The data shown in this study were plotted as the mean ± standard error of the mean (SEM) of triplicate samples and were representative of at least three separate experiments. Comparisons were made among the groups by One-way analysis of variance (ANOVA) with a Bonferroni's post-test. P value of <0.05 was considered as significant.

## Animal ethics

The experiments involving the use of animals were done in accordance with guidelines of the "Institutional Animal Ethics Committee of National Institute of Immunology" which is under the control of CPCSEA. IAEC approval number was IAEC#362 / 14.

# Results

## Autophagy inducing potential of MIP Proteins, DNA and lipids

Total proteins, genomic DNA and whole lipid fractions were obtained using the protocols described in methods section. RAW 264.7 macrophages were treated with different concentrations of these fractions for 12 h after which total cell lysate was prepared. Western blotting was performed to detect the level of lipidated LC3-II in these groups. MIP protein fraction was not able to induce autophagy in macrophages while MIP lipids and DNA resulted in nearly two-fold increase in autophagy induction (**Fig 1**).

## Isolation of LAM and analysis of its immunostimulatory property

MIP-LAM was isolated as per the protocol described in methods section. Crude LAM obtained after chloroform: methanol (2:1) extraction was further purified by passing it through sephacryl S-100 column. Fractions collected were analyzed for the presence of LAM by estimating the carbohydrate content and by ELISA using commercially available LAM specific mAbs CS-35 and CS-40 (**Fig 2A**). Fraction number 19 to 34 reacted strongly with mAbs and also had higher carbohydrate content. These fractions were pooled and further examined by SDS PAGE (**Fig 2B**). A single diffused band was observed on the gel after silver staining. The identity of this diffused band as LAM was further confirmed by Western blotting (**Fig 2C**).

Fractions 35 to 46 showed low reactivity with monoclonal antibodies, indicating low LAM content in these fractions. This could be due to the presence of other carbohydrate molecules

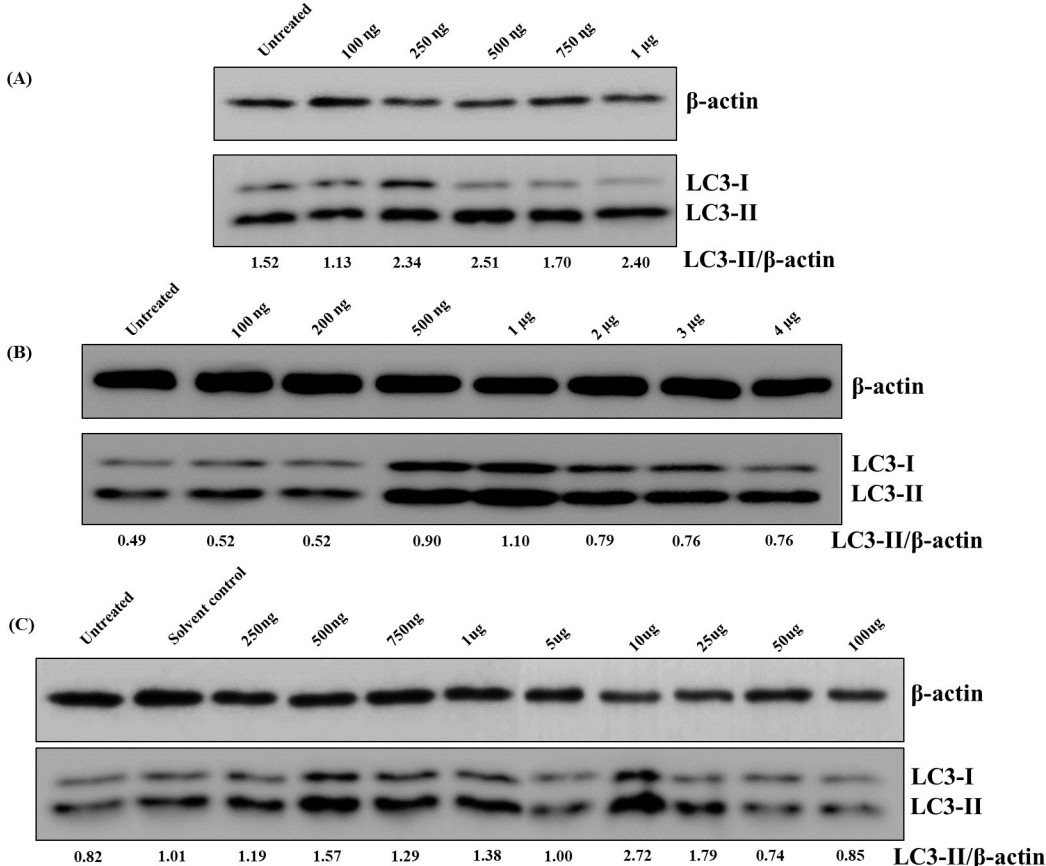

**Fig 1. MIP lipids and DNA but not proteins induce autophagy in RAW 264.7 macrophages.** RAW 264.7 macrophages were seeded (1 x 10^6 per well) and stimulated with MIP protein lysate, MIP genomic DNA and MIP whole lipid fraction for 12 h. Figure shows Western blots depicting LC3-II level in macrophages stimulated with **(A)** MIP protein lysate (concentration ranging from 100 ng to 1000 ng) **(B)** MIP genomic DNA (100 ng to 4 μg) and **(C)** MIP lipids (250 ng to 100 μg). Significant fold change in LC3-II expression with respect to unstimulated control was observed between concentrations ranging from 500 ng to 1μg of MIP-DNA and 10 μg to 25 μg of whole lipid fraction.

present along with LAM. Several components viz. lipomannan and phosphatidylinositol mannoside have been shown to be present along with LAM after phenol extraction step.

Further, immunostimulatory property of MIP-LAM was analyzed on macrophages and BMDCs and compared with that of *M.tb*-LAM. This was done by stimulating J774.1 macrophages / BMDCs with varying concentrations of MIP-LAM and *M.tb*-LAM for 24 h and supernatant was collected. The level of pro-inflammatory cytokines in culture supernatant was determined by ELISA. MIP-LAM stimulation resulted in higher amount of pro-inflammatory cytokines: TNF-α, IL-12 and IL-6 at concentrations ranging from 1 μg to 10 μg as compared to *M.tb*-LAM stimulated macrophages (**Fig 2D**) and BMDCs; very minimal level of these cytokines were observed upon stimulation with *M.tb*-LAM (**Fig 2E**). These results provide evidence of immunostimulatory property of MIP-LAM on both cell types.

## ROS and NO production in macrophages upon stimulation with MIP-LAM

To examine if MIP-LAM promote other antimicrobial defenses, ROS and NO production by MIP-LAM was studied. One group stimulated with LPS was taken as control. It was observed that ROS production was similar in LPS and MIP-LAM stimulated macrophages but NO production was very minimal in MIP-LAM group, while it was substantial in the LPS stimulated

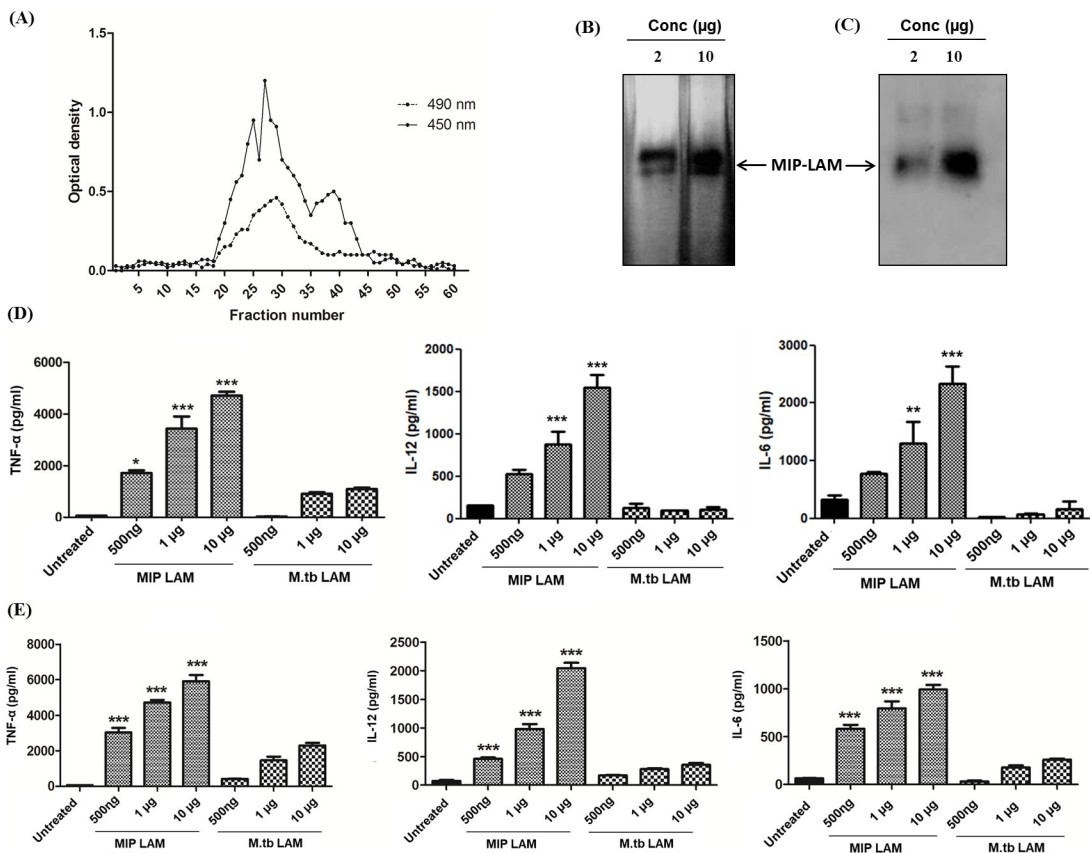

**Fig 2. Isolation and purification of LAM.** Crude LAM (10 mg/ml PBS) was passed through sephacryl S-100 column (equilibrated with PBS) at flow rate of 0.5 ml/min. Fractions of 3 ml each were collected. Carbohydrate content (dotted line, O.D. at 490 nm) and LAM content (Solid line, O.D. at 450 nm) in each fraction was determined by phenol-sulphuric acid method and by ELISA **(A)** Fractions 19 to 34 were pooled, concentrated and presence of LAM was confirmed by silver staining **(B)** and Western blotting **(C). Analysis of immunostimulatory property of MIP-LAM.** Macrophages / BMDCs were stimulated with varying concentrations of MIP-LAM / *M.tb*-LAM for 24 h. Level of TNF-α, IL-6 and IL-12 in the culture supernatant of macrophages **(D)** or BMDCs **(E)** was determined by ELISA. Data represents the mean with SEM of three independent experiments. [*: p ≤ 0.05; **: p ≤ 0.01; ***: p ≤ 0.001; comparisons were made between cytokines induced by same concentration of MIP-LAM and *M.tb*-LAM].

macrophages (**S1 Fig**). These observations provide evidence that MIP-LAM has immunostimulatory effect on macrophages. Whereas, inhibitory metabolic effect of *M.tb*-LAM on macrophages, is well established in literature.

## Autophagy induction potential of MIP-LAM

As MIP lipid fraction had shown high autophagy inducing potential and also, MIP-LAM demonstrated strong immunostimulatory property, we further analyzed its ability to induce autophagy in macrophages. Briefly, 1 x 10⁶ macrophages were seeded in 12-well plate and incubated with varying concentrations of MIP-LAM / *M.tb*-LAM for 2, 4, 8 and 12 h and lysate was prepared followed by Western blotting for LC3-II. Densitometry analysis of the LC3 blot indicated that MIP-LAM was able to induce significant level of autophagy as compared to untreated control and *M.tb*-LAM treated macrophages. Level of LC3-II was found to be higher in MIP-LAM group at all time points studied in comparison to *M.tb*-LAM group. However, at

2 h and 4 h of stimulation, maximum fold change in the LC3-II level was observed with MIP-LAM concentration of 2 μg and 5 μg. (S2 Fig).

For further stimulation experiments, 4 h time point was used. Apart from LC3 Western blot, higher autophagy induction of MIP-LAM was confirmed by puncta formation assay where similar results were observed. Average puncta formed per cell were quantified in MIP-LAM / *M.tb*-LAM stimulated macrophages. MIP-LAM resulted in dense puncta formation whereas sparse puncta were observed in *M.tb*-LAM stimulated groups (Fig 3).

### Assessment of co-localization of *M.tb* within phago-lysosomes

Since, MIP-LAM induced significantly high autophagy in macrophages; we further studied if it affects the co-localization of *M.tb* in lysosomes. Macrophages were infected with GFP expressing *M.tb*, and were subsequently stimulated with MIP-LAM or *M.tb*-LAM. MIP-LAM treatment resulted in significant increase in the co-localization of GFP expressing *M.tb* within the lysosomes (Fig 4A). *M.tb* presence inside the lysosomes was found to be 61% as compared to 24% in control unstimulated group while it was 41% in the whole lipid treated group (Fig 4B). This suggests that MIP-LAM is an important component of lipid fraction of MIP responsible for enhanced co-localization of *M.tb* within the lysosomes. However, in *M.tb*-LAM stimulated group, the co-localization was approximately 21%, which was similar to that of unstimulated control group.

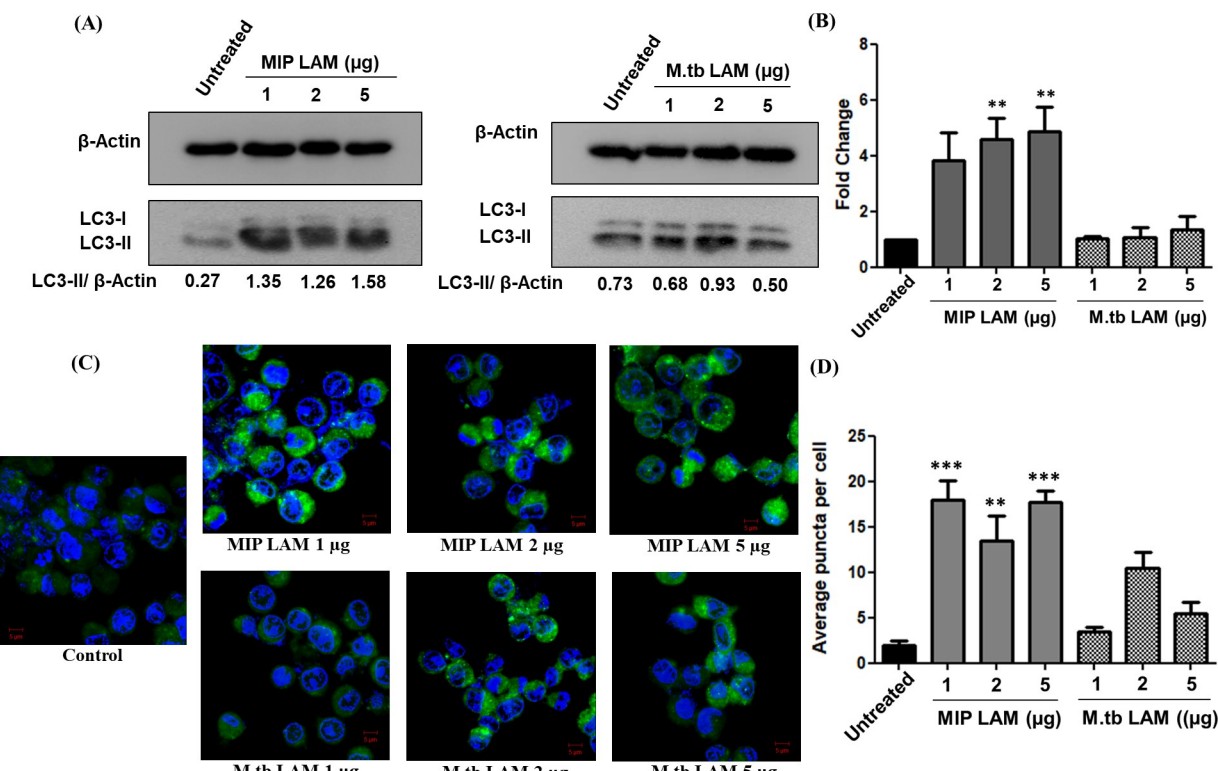

**Fig 3. MIP-LAM stimulation results in significant induction of autophagy in macrophages.** RAW 264.7 macrophages were stimulated with varying concentrations of MIP-LAM / *M.tb*-LAM for 4 h. Lysates were prepared and Western blotting was performed to analyze the expression level of LC3. (**A**) Shown are the representative blots depicting the level of lipidated LC3-II in MIP-LAM / *M.tb*-LAM stimulated macrophages. (**B**) Graph depicting the mean fold change ± range in LC3-II level in all the groups. (**C**) RAW cells stimulated with MIP-LAM / *M.tb*-LAM were analyzed for LC3 puncta formation by immunofluorescence. Shown are the merged images of GFP-LC3 (green) and DAPI (blue) taken at 63 X magnification. (**D**) Graph showing average puncta formed per cell in LAM stimulated macrophages. The scale bar represents 5 μm. **: P<0.001, ***: P<0.0001.

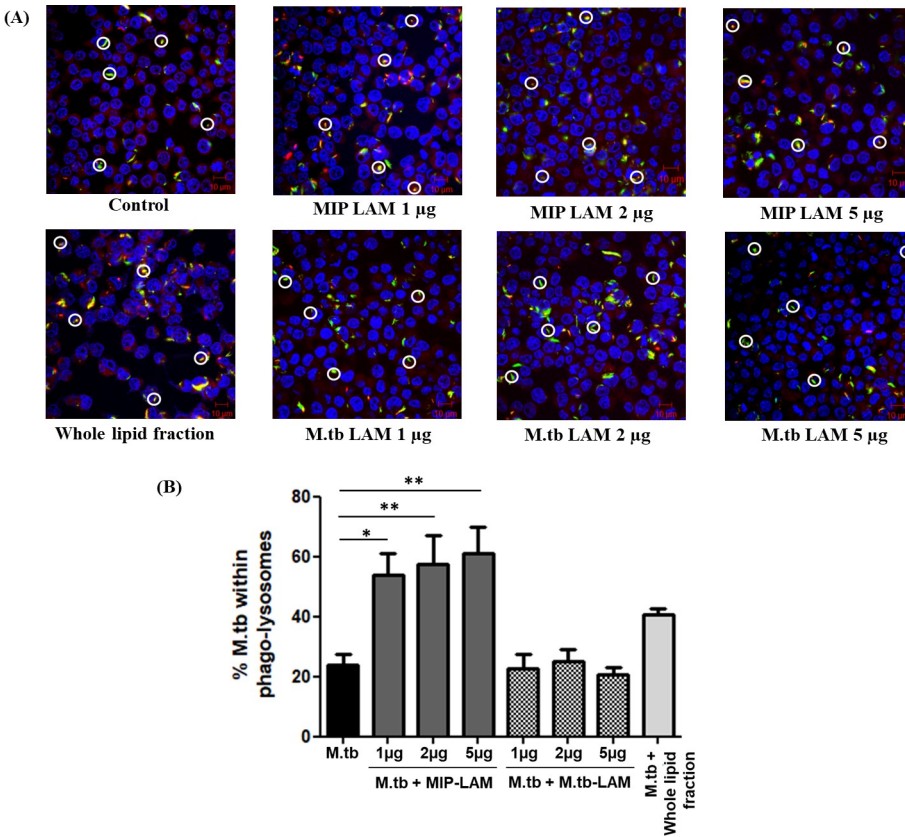

**Fig 4. Stimulation with MIP-LAM increases co-localization of *M.tb* within the phago-lysosomes.** RAW 264.7 macrophages were infected with GFP expressing *M.tb* for 4 h with subsequent stimulation with MIP-LAM / *M.tb*-LAM for another 4 h. Lysosomes were stained with LysoTracker-Red and visualised by confocal microscope. From each group, 50 fields were examined and total number of green spots (GFP expressing *M.tb* located outside the lysosomes) and of yellow spots (observed when GFP expressing *M.tb* co-localizes with LysoTracker-Red giving a yellow fluorescence) were counted. **(A)** Representative images showing co-localization of *M.tb* within the lysosomes. Yellow and white circles are the representations of *M.tb* located inside and outside of the lysosomes of the macrophages, respectively. Scale bar, 10 μm. **(B)** Graph depicting the percentage of *M.tb* co-localized within the lysosomes. *: P< 0.01, **: P<0.001.

## Effect of MIP-LAM on survival of *M.tb* in the infected macrophages

Next, we examined the effect of MIP-LAM on survival of *M.tb* in the macrophages. Macrophages were first infected with GFP expressing *M.tb* for 4 h. Subsequently, the cells were stimulated with MIP-LAM / *M.tb*-LAM for another 4 h. Thereafter, lysate was prepared and plated on hygromycin containing 7H11-agar plates. CFU were counted and percentage survival of *M.tb* was calculated. Stimulation with MIP-LAM resulted in approximately 65% decrease in *M.tb* CFU count while MIP whole lipid fraction treatment reduced the *M.tb* CFU count by 50% as compared to unstimulated control group. *M.tb*-LAM had only minimal effect where *M.tb* CFU reduction was only 15% (Fig 5).

## Discussion

Mycobacterial species induce autophagy in macrophages but the extent of induction varies with species [12]. The non-pathogenic species induce strong autophagic response in macrophages whereas pathogenic ones are known to suppress autophagy which form a part of

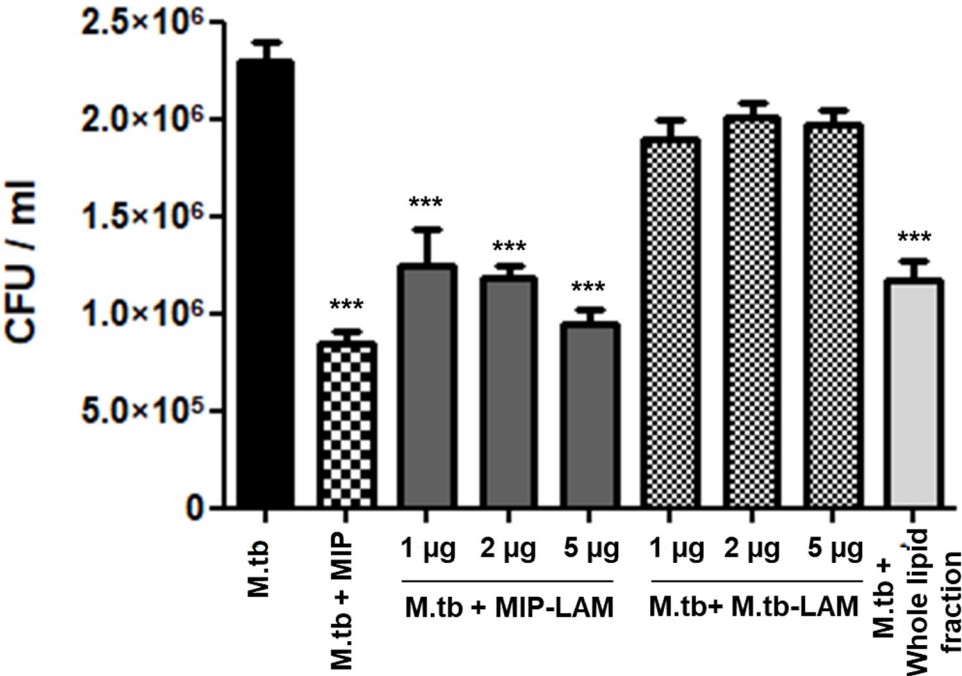

**Fig 5. MIP-LAM stimulation leads to enhanced clearance of *M.tb* from infected macrophages.** Macrophages were infected with GFP expressing *M.tb* for 4 h and subsequent stimulation with varying concentrations of MIP-LAM/ *M.tb*-LAM for 4 h. Group stimulated with MIP whole lipid fraction was taken for comparison. Cells lysate was prepared and plated (100 µl from each group) in triplicates on 7H11-agar plates. Shown here is the mean CFU count of *M.tb* from 4 independent experiments. ***: P<0.0001.

their survival strategies within the host cells. However, this suppression/ inhibition of autophagy is caused by certain mycobacterial components including lipoproteins (LpqH, LprE) and glycolipids (phosphatidyl-*myo*-inositol mannosides, lipomannans, lipoarabinomannan) which have the ability to limit host autophagy responses to the pathogens [15,21–24]. In recent past, we have reported MIP to be a potent inducer of autophagy. The main focus of this study was to determine the fractions of MIP which possess the ability to induce autophagic response. Protein fraction was incapable of inducing autophagy, whereas genomic DNA showed some autophagic response but it was less than that of lipid fraction which exhibited significant induction of autophagy (**Fig 1**). These results are consistent with previous findings where it has been shown that lipids are crucial in membrane remodeling, formation of autophagosomes and signal transduction process which lead to completion of autophagy process [25,26].

These results encouraged us to isolate LAM from MIP and to study its attributes. LAM, which is ubiquitously found in all species of mycobacteria, is a complex molecule containing a phosphatidylinositol (PI) moiety anchoring a large mannose core to the mycobacterial cell wall [27]. The pathogenic mycobacteria mainly *M.tb* and *M.leprae* possess Mannose-capped lipoarabinomannan (ManLAM); whereas, rapidly growing mycobacterial species have Arabinose-LAM [28,29]. LAM exhibits a wide range of immunomodulatory functions and is considered to be a critical factor which helps mycobacteria to modulate phagocyte functions contributing to the persistence of mycobacteria within macrophages [27,30,31]. *M.tb*-LAM has been reported to be responsible for a few distinct features of *M.tb* in phagocytic cells, inclusive of phagosome maturation arrest, autophagy inhibition, inhibition of macrophage apoptosis and limiting the production of pro-inflammatory cytokines [32].

Autophagy induction by MIP lipids co-relates to our ongoing concurrent studies where we have shown that MIP induced autophagy is mediated by the TLR2 signaling recognized by mycobacterial LAM. It is possible that MIP-LAM could be the crucial component responsible for autophagy induction in macrophages. In addition to LAM, other glycolipids including lipomannan, phosphatidyl-myo-inositol mannoside (PIM) and trehalosedimycolate (TDM), as well as certain tri-acylated lipoproteins present in *M.tb* cell-wall are also reported to interact with TLR2 leading to activation of immune cells [33].

Properties of LAM leading to activation of macrophages and other immune cells have been extensively studied. Ara-LAM from non-pathogenic (fast growing) mycobacterial species is known to be more potent than Man-LAM from pathogenic mycobacteria (slow growing particularly *M.tb*) in stimulating the macrophages to evoke secretion of cytokines [34]. Ara-LAM exhibits the ability to elicit high TNF-α production in macrophages, whereas little or negligible amount of TNF-α production is seen in response to the treatment with Man-LAM [35,36]. Besides, Ara-LAM has also displayed the capacity to trigger production of various other cytokines including IL-6, IL-8, IL-10, and IL-12 by the host cells [30,37]. Very similar to these results, we also observed that MIP-LAM isolated from MIP, which is a non-virulent mycobacterium, resulted in the production of significantly high levels of TNF-α in macrophages, whereas *M.tb*-LAM treatment was not able to induce notable TNF-α production (**Fig 2**).

*M.tb*-LAM is reported to inhibit autophagy as well as phago-lysosome fusion in macrophages. In this study, autophagy inducing ability of MIP-LAM was examined. Interestingly, stimulation of macrophages with MIP-LAM resulted in high induction of autophagy as compared to unstimulated control (**Fig 3**). MIP-LAM stimulated macrophages displayed significantly high percentage of *M.tb* co-localization in the phago-lysosomal compartments unlike that of *M.tb*-LAM (**Fig 4**). Also, MIP-LAM resulted in decreased survival of *M.tb* in infected macrophages. The percentage survival of *M.tb* in MIP-LAM stimulated groups was comparable to that of macrophages stimulated with MIP whole lipid fraction (**Fig 5**).

This study shed light on the autophagy inducing potential of different fractions of MIP, particularly LAM isolated from lipid fraction. This is a probing study, providing evidence of crucial role of MIP-LAM in clearance of *M.tb* from the host by inducing secretion of pro-inflammatory cytokines as well as by inducing autophagy. Further characterization of MIP-LAM will throw light on its composition and would elaborate its other attributes.

## Supporting information

**S1 Fig. MIP-LAM induces high ROS but minimal NO in macrophages. (A)** RAW macrophages were left unstimulated or stimulated with LPS (1 μg/mL) or MIP-LAM (5 μg/mL) for 24 h. Supernatant was collected and estimated for NO with Griess reagent. **(B)** ROS in macrophages was determined by using 2',7'-dichlorodihydrofluorescein diacetate (H$_2$DCFDA) dye. Fluorescence was determined using fluorimeter excitation at 490 nm and emission at 524 nm. H$_2$O$_2$ (100 μM) was used as a positive control. $^{***}$: P<0.0001.
(TIF)

**S2 Fig. Kinetics of LC3-II level induced by MIP-LAM/ *M.tb*-LAM.** RAW 264.7 macrophages were stimulated with various concentrations of MIP-LAM / *M.tb*-LAM for 2, 4, 8 and 12 h. Shown are the Western blots for the indicated time points.
(TIF)

## Acknowledgments

This work was supported by the Core Research Grant of the National Institute of Immunology.

## Author Contributions

**Conceptualization:** Bindu Singh, Mohd Saqib, Sangeeta Bhaskar.

**Data curation:** Bindu Singh, Mohd Saqib, Anush Chakraborty.

**Formal analysis:** Bindu Singh, Mohd Saqib, Sangeeta Bhaskar.

**Funding acquisition:** Sangeeta Bhaskar.

**Investigation:** Sangeeta Bhaskar.

**Methodology:** Bindu Singh, Mohd Saqib, Sangeeta Bhaskar.

**Project administration:** Sangeeta Bhaskar.

**Resources:** Sangeeta Bhaskar.

**Supervision:** Sangeeta Bhaskar.

**Validation:** Mohd Saqib.

**Visualization:** Bindu Singh, Mohd Saqib.

**Writing – original draft:** Bindu Singh.

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
