## [Decision Letter · Decision Letter 0]

19 Jun 2019

PONE-D-19-15439

Lipoarabinomannan from Mycobacterium indicus pranii shows immunostimulatory activity and induces autophagy in macrophages

PLOS ONE

Dear Dr. Bhaskar,

Thank you for submitting your manuscript to PLOS ONE. After careful consideration, we feel that it has merit but does not fully meet PLOS ONE’s publication criteria as it currently stands. Therefore, we invite you to submit a revised version of the manuscript that addresses the points raised by the reviewer. Several controls are missing to definitely conclude on the data. In addition, the purity of the LAM used is key to the study. Indeed, in previous reports in the litterature, activities associated to LAM actually arose from contaminants, either endogenous lipoproteins or exogenous LPS (collected during the purification steps). Authors used a phenol extraction procedure, which ensures a lipoprotein-free preparation. However, they must control that their LAM preparation does not activate TLR4 signaling.

We would appreciate receiving your revised manuscript by Aug 03 2019 11:59PM. To enhance the reproducibility of your results, we recommend that if applicable you deposit your laboratory protocols in protocols.io, where a protocol can be assigned its own identifier (DOI) such that it can be cited independently in the future. For instructions see: http://journals.plos.org/plosone/s/submission-guidelines#loc-laboratory-protocols

We look forward to receiving your revised manuscript.

Kind regards,

Jérôme Nigou

Academic Editor

PLOS ONE

**Journal Requirements**

2. Our internal editors have looked over your manuscript and determined that it is within the scope of our Autophagy and Proteostasis Call for Papers. This collection of papers is headed by a team of Guest Editors: Sharon Tooze, Fulvio Regiori and Thorsten Hoope. The Collection will encompass a diverse range of research articles from early initiation of autophagy, to understand the role other proteostasis pathways play in maintaining cellular homeostasis and the cross talk between the two.  Additional information can be found on our announcement page: https://collections.plos.org/s/autophagy-proteostasis.

If you would like your manuscript to be considered for this collection, please let us know in your cover letter and we will ensure that your paper is treated as if you were responding to this call. If you would prefer to remove your manuscript from collection consideration, please specify this in the cover letter.

3. Thank you for including your ethics statement: "The experiments involving the use of animals were done in accordance with the Institute’s Animal Ethics guidelines. IAEC approval number was IAEC#362/ 14"

Please amend your current ethics statement to include the full name of the ethics committee that approved your specific study.

For additional information about PLOS ONE submissions requirements for animal ethics, please refer to http://journals.plos.org/plosone/s/submission-guidelines#loc-animal-research  

**Comments to the Author**

1. Is the manuscript technically sound, and do the data support the conclusions?

Reviewer #1: Yes

2. Has the statistical analysis been performed appropriately and rigorously? 

Reviewer #1: Yes

3. Have the authors made all data underlying the findings in their manuscript fully available?

Reviewer #1: Yes

4. Is the manuscript presented in an intelligible fashion and written in standard English?

Reviewer #1: Yes

5. Review Comments to the Author

Reviewer #1: The manuscript submitted by Bhaskar and colleagues reports on the immunostimulatory properties of lipoarabinomannan from Mycobacterium indicus pranii (MIP-LAM). Authors showed that MIP-LAM induces autophagy and proinflammatory cytokines in murine macrophages. They found that MIP-LAM promotes acidification of Mycobacterium tuberculosis (MTB)-containing compartment and killing of the MTB in macrophages. Overall, this set of data demonstrates that MIP-LAM activates antimicrobial defenses of macrophages.

Major comments:

-It would be important to add the following controls to show that immunostimulatory properties is not due to LPS contamination and that decrease of CFU is not due to increase in cell death.

-Does MIP-LAM promote other antimicrobial defenses such as ROS and NO production?

-Authors compare MIP-LAM with MTB-LAM. It would have been interesting to compare with the LAM from another non-pathogenic mycobacteria such as M.smegmatis.

-Regarding autophagy assays: it would be important to do a kinetic with one concentration of LAM (compare MIP-LAM and MTB-LAM) and to measure autophagic flux by adding lysosome inhibitors (see Klionsky et al. Guidelines. Autophagy. 2016).

-The ratios of LC3-II/actin need to be measured and display at the bottom of the blots.

-Some references are missing: l72, l78, l424. L424: which lipoproteins and glycolipids? Please add reference “Sui et al. 2011. J Proteome Res” for lipoarabinomannan and autophagy and “Bah et al. 2016. Front Cell Infect Microbiol” for lipoglycan and autophagy.

-Please add a paragraph in materials & methods for LC3 westernblot.

-l277: authors indicate that lysotracker was added after fixing. The standard protocol indicates an incubation with living cells then fixing: https://www.thermofisher.com/order/catalog/product/L7528. I am not sure that the assay was done properly. Please clarify and give a reference for the protocol (from another laboratory).

-l303: please be more specific: fold of increase, concentration.

-l306: please indicate incubation time.

-Figure1: please label properly the figure, what are the bottom and upper blots?

Minor comments:

-Please spell “TE” out (l141)

-l269: please change “flour” to “fluor”

-l272: please change “co-localization of M.tb in lysosomes of LAM stimulated macrophages” to “co-localization of M.tb with lysosomes in LAM-stimulated macrophages”.

-l301: please change “LC3” to “LC3-II”

-Figure 3 legend: indicate scale bar, change ug; in panel D remove one (.

-Figure 4 legend: please change “no” to “number” and add “of” before “yellow”; indicate scale bar; what are those white circles?

-l412: remove (A)

6. PLOS authors have the option to publish the peer review history of their article (what does this mean?). If published, this will include your full peer review and any attached files.

Reviewer #1: No

---

## [Author Response · Author response to Decision Letter 0]

20 Aug 2019

Journal Requirements

Response: For the current submission, we have ensured that the manuscript meets PLOS ONE’s requirements.

2. Our internal editors have looked over your manuscript and determined that it is within the scope of our Autophagy and Proteostasis Call for Papers. This collection of papers is headed by a team of Guest Editors: Sharon Tooze, Fulvio Regiori and Thorsten Hoope. The Collection will encompass a diverse range of research articles from early initiation of autophagy, to understand the role other proteostasis pathways play in maintaining cellular homeostasis and the cross talk between the two. Additional information can be found on our announcement page: https://collections.plos.org/s/autophagy-proteostasis.

If you would like your manuscript to be considered for this collection, please let us know in your cover letter and we will ensure that your paper is treated as if you were responding to this call. If you would prefer to remove your manuscript from collection consideration, please specify this in the cover letter.

Response: Yes, please consider this manuscript for the collection- “Autophagy and Proteostasis”. We have mentioned the same in our cover letter.

3. Thank you for including your ethics statement: "The experiments involving the use of animals were done in accordance with the Institute’s Animal Ethics guidelines. IAEC approval number was IAEC#362/ 14"

Please amend your current ethics statement to include the full name of the ethics committee that approved your specific study.

For additional information about PLOS ONE submissions requirements for animal ethics, please refer to http://journals.plos.org/plosone/s/submission-guidelines#loc-animal-research.

Response: We have amended the ethics statement in the revised manuscript. The full name of the animal ethics committee is “Institutional Animal Ethics Committee of National Institute of Immunology” under control of CPCSEA. We will also amend the “Ethics Statement” in submission form.

Response:This study doesn’t involve any repository information. All relevant data has been provided in the manuscript and there is no additional data to upload. Please change our Data availability statement.

Comments to the Author

1. Is the manuscript technically sound, and do the data support the conclusions?

Reviewer #1: Yes

2. Has the statistical analysis been performed appropriately and rigorously? 

Reviewer #1: Yes

3. Have the authors made all data underlying the findings in their manuscript fully available?

Reviewer #1: Yes

4. Is the manuscript presented in an intelligible fashion and written in standard English?

Reviewer #1: Yes

5. Review Comments to the Author

Reviewer #1: The manuscript submitted by Bhaskar and colleagues reports on the immunostimulatory properties of lipoarabinomannan from Mycobacterium indicus pranii (MIP-LAM). Authors showed that MIP-LAM induces autophagy and proinflammatory cytokines in murine macrophages. They found that MIP-LAM promotes acidification of Mycobacterium tuberculosis (M.tb)-containing compartment and killing of the M.tb in macrophages. Overall, this set of data demonstrates that MIP-LAM activates antimicrobial defenses of macrophages.

Major comments:

-It would be important to add the following controls to show that immunostimulatory properties is not due to LPS contamination and that decrease of CFU is not due to increase in cell death.

Response:LPS is immune-stimulatory component of gram-negative bacteria while LAM is an antigenic glycolipid component of genus Mycobacterium. It is reported that Mycobacterial antigens generate different metabolic responses in macrophages as compared to gram-negative effectors. Further, there was no chance of contamination of MIP culture with any other gram-negative bacteria. All steps were done in sterile condition with proper quality control checks.

As suggested, we analyzed LPS as control also while examining the ROS and NO production by MIP-LAM. It was observed that ROS production was similar in LPS and MIP-LAM stimulated macrophages but NO production was very minimal in MIP-LAM group, while it was substantial in the LPS stimulated macrophages {Fig 1 (A & B)}. These observations also provide indirect evidence that MIP-LAM is not contaminated with LPS.

Decrease in CFU of M.tb was due to increased phago-lysosome fusion in ‘M.tb+MIP-LAM’ group (Figure 4 of Manuscript) which resulted in enhanced clearance of M.tb from the infected macrophages.

-Does MIP-LAM promote other antimicrobial defenses such as ROS and NO production?

Response: As suggested, we examined the ROS and NO production by MIP-LAM. One group stimulated with LPS was taken as control. It was observed that ROS production was similar in LPS and MIP-LAM stimulated macrophages but NO production was very minimal in MIP-LAM group, while it was substantial in the LPS stimulated macrophages {Figure 1 (A & B)}. These observations provide evidence that MIP-LAM has immunostimulatory effect on macrophages. Whereas, inhibitory metabolic effect of M.tb-LAM on macrophages, is well established in literature.

A)

B)

Fig 1: (A) RAW macrophages were left unstimulated or stimulated with LPS (1 µg/mL) or MIP LAM (5 µg/mL) for 24 hours. Supernatant was collected and estimated for NO with Griess reagent. (B) Macrophages were incubated with 20 µM 2',7'-dichlorodihydrofluorescein diacetate (H2DCFDA) dye for 45 min at 37 �C under dark conditions. Cells were then washed and stimulated with LPS (1 µg/mL) / MIP-LAM (5 µg/mL). Fluorescence was determined using fluorimeter excitation at 490 nm and emission at 524 nm. H2O2 (100 µM) was used as a positive control. 

-Authors compare MIP-LAM with MTB-LAM. It would have been interesting to compare with the LAM from another non-pathogenic mycobacteria such as M.smegmatis.

Response: As MIP was found to be potent inducer of autophagy in previous study by our group hence, aim of this study was to further examine the component/s of MIP responsible for autophagy induction. Interestingly, we found that MIP lipids and DNA were able to induce autophagy and not the protein fraction. As LAM is one of the crucial component of mycobacterial cell wall lipids possessing immunomodulatory activity; we isolated LAM from MIP and did a comparative study with M.tb-LAM, as idea was to compare it with pathogenic mycobacteria. 

LAM from non-pathogenic mycobacteria including M.smegmatis had already been reported to have immunomodulatory properties (Das et al., 2015). In future studies where we will further characterize MIP-LAM, we would also add M.smegmatis LAM as a control.

-Regarding autophagy assays: it would be important to do a kinetic with one concentration of LAM (compare MIP-LAM and MTB-LAM) and to measure autophagic flux by adding lysosome inhibitors (see Klionsky et al. Guidelines. Autophagy. 2016).

Response: In initial standardization experiments kinetics study was done with MIP-LAM /M.tb- LAM where macrophages were stimulated for 2/4/8/12 h. Shown below are the Western blots of lysate prepared at different time points (Figure 2). LC3-II levels were found to be higher in MIP-LAM group at all time points studied as compared to M.tb-LAM group.

In our previous published study where we examined the autophagy inducing potential of whole MIP and M.tb, it was observed that both MIP and M.tb induced autophagy to the similar extent as observed from the levels of LC3-II expression. Hence, we further analyzed the autophagic flux and observed that MIP maintained the complete autophagic flux while M.tb inhibited the fusion of autophagosome with lysosome. But in the present study there was significantly higher induction of LC3-II by MIP-LAM as compared to M.tb-LAM at all the concentrations studied (Figure 3 of manuscript). Similarly, significantly higher numbers of puncta per cell were observed in the MIP-LAM stimulated group as compared to M.tb-LAM group which provides evidence of higher autophagy induction by MIP-LAM as compared to M.tb-LAM. Further, complete autophagic flux in the MIP-LAM stimulated group was confirmed by phago-lysosome fusion study where significantly higher numbers of M.tb bacilli were observed in the lysosome compartment in MIP-LAM stimulated group as compared to M.tb-LAM group (Figure 4 of manuscript).

Fig 2. Macrophages were stimulated with varying concentration of MIP-LAM/ M.tb-LAM for 2, 4, 8 and 12 h. Western blotting for LC3-II was done and density was quantified using Image J software. Shown here are the blots depicting LC3-II levels in MIP-LAM / M.tb-LAM stimulated macrophages.

-The ratios of LC3-II/actin need to be measured and display at the bottom of the blots.

Response: The ratios of LC3-II/actin have been measured and displayed at the bottom of the blots.

-Some references are missing: l72, l78, l424. L424: which lipoproteins and glycolipids? Please add reference “Sui et al. 2011. J Proteome Res” for lipoarabinomannan and autophagy and “Bah et al. 2016. Front Cell Infect Microbiol” for lipoglycan and autophagy.

Response: We are sorry for the mistake. The missing references have been included in the revised manuscript along with above mentioned references.

-Please add a paragraph in materials & methods for LC3 western blot.

Response: In the materials & methods section of the revised manuscript, a paragraph on LC3 western blot has been added.

-l277: authors indicate that lysotracker was added after fixing. The standard protocol indicates an incubation with living cells then fixing: https://www.thermofisher.com/order/catalog/product/L7528. I am not sure that the assay was done properly. Please clarify and give a reference for the protocol (from another laboratory).

Response: We are sorry for this. It was mistakenly written in methods. The incubation with lysotracker was done on live cells only. The cells were fixed thereafter. We have corrected the same in the manuscript.

-l303: please be more specific: fold of increase, concentration.

Response: We are sorry for the confusion. We have mentioned the autophagy induction by lipids and DNA in the form of fold increase as suggested.

-l306: please indicate incubation time.

Response: We have indicated the incubation time (12 h) in the manuscript.

-Figure1: please label properly the figure, what are the bottom and upper blots?

Response: Figure 1 has been labeled accordingly. These are representative blots showing LC3 expression, where upper blot represents MIP protein stimulated samples, the middle one represents MIP DNA stimulated samples and the lower one is for MIP lipid stimulated groups.

Minor comments:

-Please spell “TE” out (l141)

Response: “TE” has been spelled out in the revised manuscript.

-l269: please change “flour” to “fluor”

Response: In line 269, “flour” has been changed to “fluor”.

-l272: please change “co-localization of M.tb in lysosomes of LAM stimulated macrophages” to “co-localization of M.tb with lysosomes in LAM-stimulated macrophages”.

Response: The above heading has been accordingly modified in the revised manuscript.

-l301: please change “LC3” to “LC3-II”

Response: “LC3” has been modified to “LC3-II”.

-Figure 3 legend: indicate scale bar, change ug; in panel D remove one (.

Response: In Figure 3 legend of the revised manuscript, above mentioned changes have been made.

-Figure 4 legend: please change “no” to “number” and add “of” before “yellow”; indicate scale bar; what are those white circles?

Response: In figure 4 legend, the required modifications have been done as well as scale bars have been indicated. White circles shown in the images were to highlight the localization of GFP expressing M.tb, either inside (yellow spots) or outside (green spots) the lysosomes in various experimental groups.

To avoid confusion, we have now changed the color used for highlighting the bacilli. Now, yellow circles represent bacilli inside lysosomes and white circles represent bacilli outside the lysosomes. Further, we have highlighted only few bacilli (n=5) using circles as these were many in number and if all bacilli would have been marked with circles then the images will appear cluttered.

-l412: remove (A)

Response: (A) has been removed from the figure legend.

---

## [Decision Letter · Decision Letter 1]

22 Aug 2019

PONE-D-19-15439R1

Lipoarabinomannan from Mycobacterium indicus pranii shows immunostimulatory activity and induces autophagy in macrophages

PLOS ONE

Dear Dr. Bhaskar,

Thank you for submitting your manuscript to PLOS ONE. After careful consideration, we feel that it has merit but does not fully meet PLOS ONE’s publication criteria as it currently stands. Therefore, we invite you to submit a revised version of the manuscript that addresses the points raised during the review process.

Additional experiments performed for the revision are important and should be included, described and analyzed in the revised manuscript.

We would appreciate receiving your revised manuscript by Oct 06 2019 11:59PM. To enhance the reproducibility of your results, we recommend that if applicable you deposit your laboratory protocols in protocols.io, where a protocol can be assigned its own identifier (DOI) such that it can be cited independently in the future. For instructions see: http://journals.plos.org/plosone/s/submission-guidelines#loc-laboratory-protocols

We look forward to receiving your revised manuscript.

Kind regards,

Jérôme Nigou

Academic Editor

PLOS ONE

Reviewers' comments:

Reviewer's Responses to Questions

**Comments to the Author**

1. If the authors have adequately addressed your comments raised in a previous round of review and you feel that this manuscript is now acceptable for publication, you may indicate that here to bypass the “Comments to the Author” section, enter your conflict of interest statement in the “Confidential to Editor” section, and submit your "Accept" recommendation.

Reviewer #1: All comments have been addressed

2. Is the manuscript technically sound, and do the data support the conclusions?

Reviewer #1: Yes

3. Has the statistical analysis been performed appropriately and rigorously? 

Reviewer #1: Yes

4. Have the authors made all data underlying the findings in their manuscript fully available?

Reviewer #1: Yes

5. Is the manuscript presented in an intelligible fashion and written in standard English?

Reviewer #1: Yes

6. Review Comments to the Author

Reviewer #1: Thank you for submitting a revised version of your manuscript that include additional experiments. Fig1 A & B for ROS and NO production as well as Fig2 LC3-II kinetics presented in the rebuttal letter should be included and commented in a revised manuscript. Do not forget to add in material and methods: ROS and NO production assays. Important: Legends for Fig1 (A) and (B) have been inverted, please correct.

7. PLOS authors have the option to publish the peer review history of their article (what does this mean?). If published, this will include your full peer review and any attached files.

Reviewer #1: No

---

## [Author Response · Author response to Decision Letter 1]

6 Oct 2019

Response: As per the suggestions, we have included the additional experiments and their results in the revised manuscript.

---

## [Editor Report · Decision Letter 2]

9 Oct 2019

Lipoarabinomannan from Mycobacterium indicus pranii shows immunostimulatory activity and induces autophagy in macrophages

PONE-D-19-15439R2

Dear Dr. Bhaskar,

We are pleased to inform you that your manuscript has been judged scientifically suitable for publication and will be formally accepted for publication once it complies with all outstanding technical requirements.

With kind regards,

Jérôme Nigou

Academic Editor

PLOS ONE
---

## [Editor Report · Acceptance letter]

14 Oct 2019

PONE-D-19-15439R2 

Lipoarabinomannan from Mycobacterium indicus pranii shows immunostimulatory activity and induces autophagy in macrophages 

Dear Dr. Bhaskar:

I am pleased to inform you that your manuscript has been deemed suitable for publication in PLOS ONE. Congratulations! Your manuscript is now with our production department. 

With kind regards,

on behalf of

Dr. Jérôme Nigou 

Academic Editor

PLOS ONE